# Epidemiology of Amyloidosis and Genetic Pathways to Diagnosis and Typing

**Kari Hemminki [1,2,\*] and Asta Försti [3,4]**

1. Biomedical Center, Faculty of Medicine and Biomedical Center in Pilsen, Charles University in Prague, 30605 Pilsen, Czech Republic
2. Division of Cancer Epidemiology, German Cancer Research Center (DKFZ), Im Neuenheimer Feld 580, 69120 Heidelberg, Germany
3. Hopp Children's Cancer Center (KiTZ), 69120 Heidelberg, Germany; a.foersti@dkfz.de
4. Division of Pediatric Neurooncology, German Cancer Research Center (DKFZ), German Cancer Consortium (DKTK), 69120 Heidelberg, Germany
* Correspondence: k.hemminki@dkfz.de

**Abstract:** We reviewed our studies on epidemiology and germline genetics of amyloidosis. In epidemiology, we considered both hereditary and non-hereditary amyloidosis. As the source of data, we used the nationwide Swedish hospital discharge register. We estimated the incidence of hereditary ATTR amyloidosis, for which Sweden is a global endemic area, at 2/million. Surprisingly, the disease was also endemic within Sweden; the incidence in the province with the highest incidence was 100 times higher than in the rest of Sweden. Risk of non-Hodgkin lymphoma increased five-fold in the affected individuals. Among non-hereditary amyloidosis, the incidence for AL amyloidosis (abbreviated as AL) was estimated at 3.2/million, with a median survival time of 3 years. Secondary systemic amyloidosis (most likely AA amyloidosis) showed an incidence of 1.15/million for combined sexes. The female rate was two times higher than the male rate, probably relating to the higher female prevalence of rheumatoid arthritis. The median survival time was 4 years. We also identified patients who likely had familial autoinflammatory disease, characterized by early onset and immigrant background from the Eastern Mediterranean area. Young Syrian descendants had the highest incidence rate, which was over 500 times higher than that in individuals with Swedish parents. Germline genetics focused on AL on which we carried out a genome-wide association study (GWAS) in three AL cohorts ($N = 1129$) from Germany, UK, and Italy. Single nucleotide polymorphisms (SNPs) at 10 loci showed evidence of an association at $p < 10^{-5}$; some of these were previously documented to influence multiple myeloma (MM) risk, including the SNP at the IRF4 binding site. In AL, SNP rs9344 at the splice site of cyclin D1, influencing translocation (11;14), reached the highest significance, $p = 7.80 \times 10^{-11}$; the SNP was only marginally significant in MM. The locus close to gene SMARCD3, involved in chromatin remodeling, was also significant. These data provide evidence for common genetic susceptibility to AL and MM. We continued by analyzing genetic associations in nine clinical profiles, characterized by organ involvement or Ig profiles. The light chain only (LCO) profile associated with the SNP at the splice site of cyclin D1 with $p = 1.99 \times 10^{-12}$. Even for the other profiles, distinct genetic associations were found. It was concluded that the strong association of rs9344 with LCO and t(11;14) amyloidosis offer attractive mechanistic clues to AL causation. Mendelian randomization analysis identified associations of AL with increased blood monocyte counts and the tumor necrosis factor receptor superfamily member 17 (TNFRSF17 alias BCMA) protein. Two other associations with the TNFRSF members were found. We discuss the corollaries of the findings with the recent success of treating t(11;14) AL with a novel drug venetoclax, and the application of BCMA as the common target of plasma cell immunotherapies.

**Keywords:** incidence; hereditary amyloidosis; periodic fever syndrome; translocation (11;14); cyclin D1

**Epilogue**

I, Kari Hemminki, began my medical studies at the University of Helsinki in the late 1960s, while working on my PhD in medical chemistry as a pastime. Another student, Peter Maury, worked in the same department on his topic, which he believed might be related to a rare disease: amyloidosis. After we finished our theses, our roads departed; he became a clinician and I remained in preclinical areas. Peter (CP Maury) made a career, primarily in various forms of hereditary amyloidosis. He headed a team that characterized the gelsolin gene defect, causing the Finnish hereditary amyloidosis.

My next exposure to amyloidosis was through epidemiology at the Karolinska Institute and through genetics at the German Cancer Research Center, but these episodes will be described in more detail below.

## 1. Introduction

Readers of this issue will find an introduction into amyloidosis, and Professor Merlini's contribution to the area in many papers and in our citations. We review our studies on amyloidosis epidemiology and germline genetics. Our focus is on immunoglobulin light chain (AL) amyloidosis (abbreviated here 'AL'), with the exception that, in the epidemiology section, different non-hereditary amyloidosis types and one hereditary amyloidosis, hereditary ATTR amyloidosis (also known as familial amyloidotic polyneuropathy, FAP), are discussed. The convention for amyloidosis nomenclature for abbreviations starts with an 'A' for amyloid, adding a suffix designating the precursor protein, e.g., AL is amyloid derived from the immunoglobulin light chain precursor.

## 2. Epidemiology of Amyloidosis and Related Cancers

Data on the incidence of amyloidosis is not abundant, which is typical of diseases for which no exact international classification of disease (ICD) codes have been available. In the case of amyloidosis, this is probably a reflection of the rarity, the diagnostic complexity, and heterogeneity of the condition. In 1992, Robert Kyle and coworkers wrote, " . . . we are unaware of any published studies on the incidence or prevalence of AL . . . " [1]. In that paper, they gave the incidence for AL in a U.S. county (Olmsted) as nine per million person–years, but the rate was based on only 21 patients. However, this rate has been widely cited ever since, because, apparently, no other incidence data have become available. In the present study, we report on the incidence of amyloidosis based on nationwide hospitalizations in Sweden, covering both inpatients and outpatients.

### 2.1. Non-Hereditary Amyloidosis

AL is a well-defined systemic amyloidosis, and other types include reactive AA (serum amyloid A as the precursor) amyloidosis, and transthyretin (ATTR, also called senile systemic) amyloidosis [2–4]. AA amyloidosis is a secondary condition in response to chronic inflammation or even cancer; e.g., some 20% of rheumatoid arthritis patients have shown some amyloid accumulation in the course of their disease [4,5]. Better control of inflammatory conditions has been reported to reduce the incidence of AA amyloidosis [4,6]. There are two forms of ATTR amyloidosis, wild type ATTR amyloidosis (ATTRwt) with a normal TTR protein, and hereditary ATTR amyloidosis (ATTRv) with a mutated TTR (discussed in the next section). ATTRwt is a disease of old men and primarily presents as heart failure [4].

In a Swedish nationwide study from 2001 to 2008, we estimated the incidence of secondary systemic amyloidosis (ICD-10 code E85.3, most likely AA amyloidosis) at 1.15/million for combined sexes [7]. The female rate was two times higher than the male rate, probably relating to the higher female prevalence of rheumatoid arthritis. The median survival time was 4 years. The incidence estimate from the UK was similar and the recent survival was estimated at 5.3 years [6,8].

For the other types of amyloidosis, no specific ICD codes were available. However, the incidence for AL was extrapolated to be 3.2/million, with a median survival time

of 3 years [7]. Kyle and coworkers updated their study for the Olmsted County to an incidence of 12/million [9]. Another U.S. study observed an increasing time-dependent incidence and gave a figure of 14/million for 2015 [10]. The most recent UK survival estimate was 4.3 years and the survival had continuously improved [6].

For ATTRwt amyloidosis, we could not estimate incidence, but considered it a part of the unaccounted proportion of 3/million [7]. Among UK Amyloidosis Centre patients, ATTRwt amyloidosis cases were about half of those with AL in the most recent period, and presented with a median survival of 3.3 years [6]. However the incidence estimates for ATTRwt amyloidosis will raise when its contribution to heart failure is recognized [4].

Using nationwide Swedish data, we analyzed risk of cancer in patients diagnosed with non-hereditary amyloidosis. The type of amyloidosis was estimated based on the medication used for amyloidosis treatment [11]. As a novel association, squamous cell skin cancer was associated with AA amyloidosis with a relative risk of 17 in those who received rheumatoid arthritis medication. An increased risk of non-Hodgkin lymphoma was associated with ATTRwt amyloidosis. The known association between AL amyloidosis and multiple myeloma (MM) was also observed and the relative risk was over 30.

In addition to plasma cell dyscrasias and cancer, few other diseases are known to be associated with AL. We found a surprising lead when we studied Swedish hospital discharge records for diseases that appeared to be common in monoclonal gammopathy of unknown significance (MGUS) patients. Our attention was attracted to two diseases that were apparently completely unrelated to AL as these were common eye diseases, senile cataract and glaucoma. We decided to carry out a formal study, not only on MGUS, but also including related plasma cell dyscrasias AL, MM, and Waldenström macroglobuline-mia [12]. Risk of senile cataract and glaucoma was significantly increased in all of these, but the highest risks (standardized incidence ratios) were found in AL patients: 2.31 for senile cataract and 2.18 for glaucoma. The reason for these associations, after any plasma cell dyscrasia, was suggested to be an M-protein related increase in blood viscosity, disturbing protein structure of the lens of the eye which is exquisitely sensitive to protein aggregation; ambient protein concentration of the lens is the highest of any tissue and lens proteins are extremely long-lived [13]. Specifically for AL, amyloid aggregates could interfere with lens structures and eye fluid drainage systems. According to earlier studies, AL patients had a particularly high risk (43.75) for 'glaucoma secondary to eye inflammation', probably because of amyloids causing chronic irritation of the eye [14,15].

### 2.2. Hereditary Amyloidosis

Using Swedish hospital discharge data, we identified 221 patients with ATTRv amyloi-dosis, giving an incidence of 2/million [16]. The disease is well-known in Sweden, which is, together with Portugal and Japan, an endemic area for this disease [17]. Surprisingly, our geographic survey of the cases showed that the disease was also highly endemic in Sweden, despite large population movements from the rural to urban areas. The incidence in the province with the highest incidence was 100 times higher than in the rest of Sweden. In a separate study, we assessed cancer incidence in the population diagnosed with ATTRv amyloidosis [18]. Risk of non-Hodgkin lymphoma increased five-fold, including the diffuse large B cell type, with a six-fold increase.

A common ATTRv mutation in endemic areas is Val30Met, which leads to early onset disease (20–40 years) in Portugal and Japan, but for unknown reasons, leads to later onset in Sweden. Carriers of these mutations have now been identified in non-endemic areas [19]. The disease is of late onset, but the neurological features resemble those seen in patients from endemic areas.

The title of the above paper on hereditary ATTR amyloidosis was "Incidence of hereditary amyloidosis and autoinflammatory diseases in Sweden: endemic and imported diseases" [16]. While analyzing ATTRv amyloidosis (for which the ICD code is termed 'neuropathic heredofamilial amyloidosis'), we paid attention to the related ICD code 'non-neuropathic heredofamilial amyloidosis', which included familial autoinflammatory

diseases. While these diseases have been historically unknown in Sweden, and we found no literature of diagnosed patients in the country, we decided to have a closer examination of this disease group.

Hospital discharge data showed that the median diagnostic age in patients with non-neuropathic heredofamilial amyloidosis was very different from that of ATTRv amyloidosis patients (Figure 1). The early onset of these conditions potentially identified them as familial autoinflammatory diseases. This was further supported by the origin of the patients; 98% were immigrants, most of whom were from the Eastern Mediterranean area. Young Syrian descendants had the highest incidence rate, which was over 500 times higher than that in individuals with Swedish parents. Thus, it was likely that the patients suffered from hereditary periodic fever syndrome [20–22]. Yet the highest incidence in Syrian immigrants (109 per million) was far below the rates in endemic areas, estimated to be as high as 1000 per million [23]. Because of the periodic high fever, the patients additionally have an increased risk of AA amyloidosis.

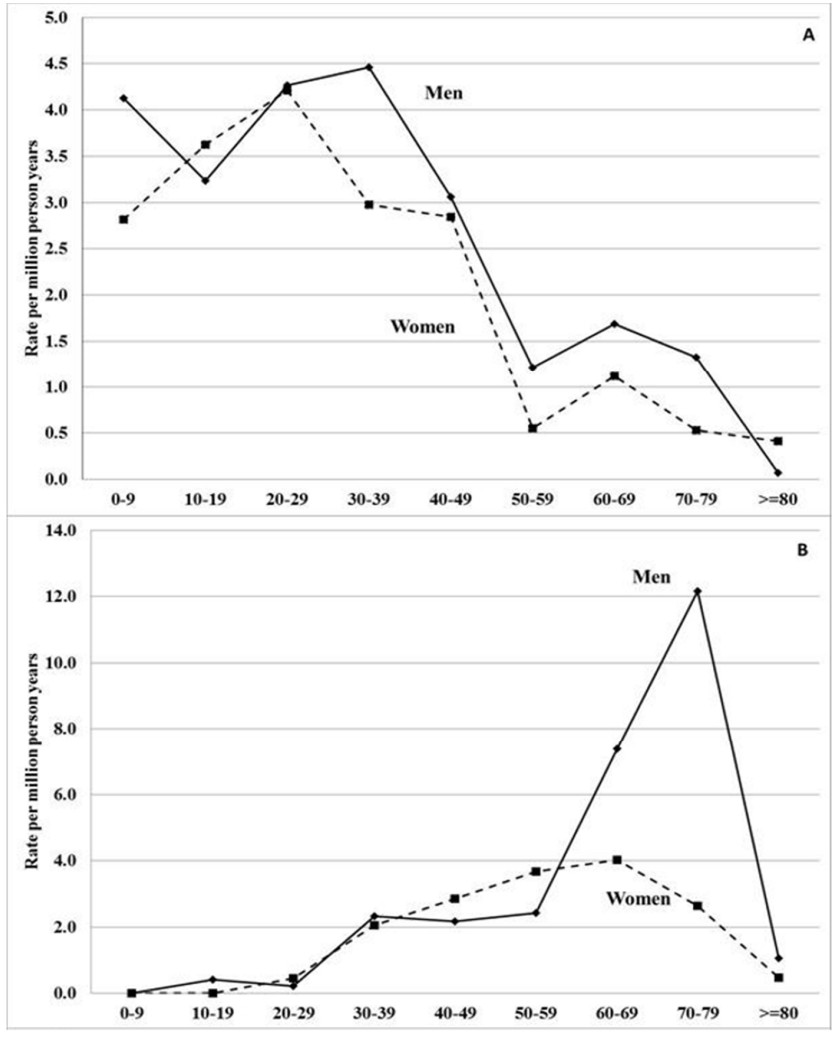

**Figure 1.** Age-specific incidence rates (per million person–years) for amyloidosis subtypes in Sweden (1997–2008), (**A**) 'non-neuropathic heredofamilial amyloidosis' or autoinflammatory disease (E85.0) and (**B**) ATTRv amyloidosis (familial amyloidotic polyneuropathy, E85.1). Modified from reference [16].

### 3. Germline Genetics of AL

Our first genetic study was published in 2014 and it was based on the hypothesis that AL may share genetic risks factors with multiple myeloma (MM) [24]. At the time, genome-wide association studies (GWASs) identified seven genetic loci for MM, and we tested these on samples from 443 AL patients. All associations were nominally significant at $p < 0.05$ and the risk alleles were identical [24]. A year earlier, we had described an association between a single nucleotide polymorphism (SNP) in the splice site of cyclin D1 (CCND1) c.870G4A (rs9344, also known as rs603965) and risk of translocation (t) t(11;14) in MM [25]. This was the first time a genetic basis for a chromosomal aberration was described. We also tested this association in AL, for which genotype and FISH data were available on 329 samples. The translocation was detected in 190 patients (58%, a much higher proportion than in MM) and the association (OR of 1.81) with SNP rs9344 was highly significant ($p = 1.5 \times 10^{-7}$). The magnitude of the association was similar in AL and MM, even though the translocation is three times more prevalent in AL. The implication is that the etiological contribution of rs9344 was three times higher in AL compared to MM.

In the era of low-risk germline genetics, the above sample size was modest, and with the help of Stefan Schönland, we were able to expand our sample to 1229 AL patients from Germany, UK, and Italy, and to 7526 healthy local controls [26]. SNPs at 10 loci showed evidence of an association at $p < 10^{-5}$ with homogeneity of results from the three sample sets (Figure 2). Some of these were shared with MM, including SNP rs4487645 at the IRF4 binding site with a genome-wide significance. The transcription factor IRF4 is an important hematopoietic regulator and it is associated with survival in MM [27,28]. With this sample size, the association with SNP rs9344 at the splice site of cyclin D1 reached a *p*-value of $7.80 \times 10^{-11}$, while in MM, the *p*-value was barely significant (0.04). SNP rs79419269 close to gene SMARCD3, involved in chromatin remodeling, was also significant ($p = 5.2 \times 10^{-8}$). One of the 10 loci marked TNFRSF13B (alias TACI), which is expressed in mature B cells and plasma cells, and discussed later, as it comes up in other analyses [29]. A separate analysis of German data on 380 patients with identified t(11;14) showed that the risk of cyclin D1 splice site SNP rs9344 was only seen in translocation (and IgH) positive cases (Table 1). The results provided evidence for common genetic susceptibility to AL and MM, but cyclin D1 appeared to be a more prominent driver in AL than in MM through its effect on t(11;14).

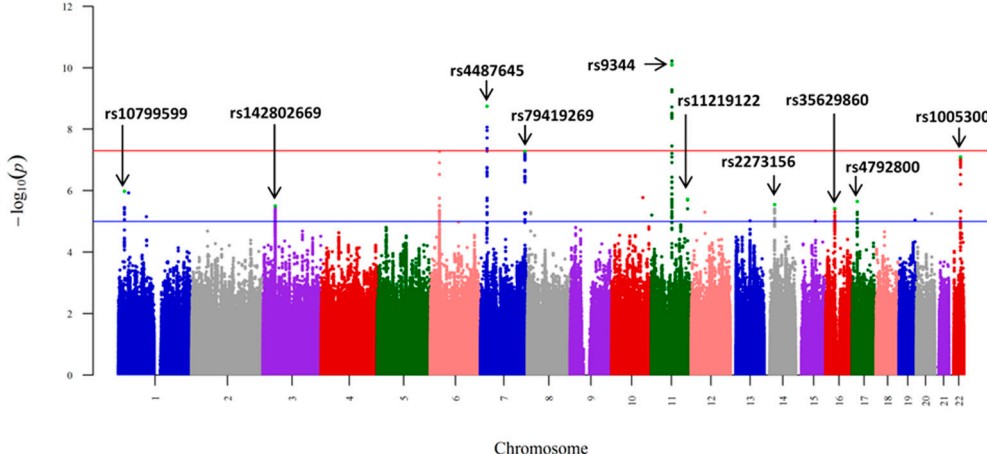

**Figure 2.** Manhattan plot on meta-analysis of the 1230 AL-amyloidosis cases and 7589 regionally matched controls. SNP IDs (rs numbers) are shown for 10 SNPs, in peaks with *p*-value below $10^{-5}$. The x-axis shows the chromosomal position and the y-axis is the significance (–$\log_{10} p$; 2-tailed) of association derived by logistic regression. The red horizontal line represents the genome-wide significance thresholds of $p = 5.0 \times 10^{-8}$ and the blue horizontal line represents the suggestive significance thresholds of $p = 1.0 \times 10^{-5}$. From reference [26] with permission.

**Table 1.** Association of SNP rs9344 with AL (*N* = 380) overall and by cytogenetic subtype (reference [22]).

| Cytogenetic Subtypes | RAF [a] | OR (95% CI) [b] | *p* |
|---|---|---|---|
| AL amyloidosis | 0.56 | 1.39 (1.22–1.60) | $2.00 \times 10^{-6}$ |
| t11;14 positive | 0.55 | 1.70 (1.38–2.09) | $4.70 \times 10^{-7}$ |
| IgH positive | 0.55 | 1.50 (1.26–1.79) | $6.60 \times 10^{-6}$ |
| t11;14 negative | 0.54 | 1.14 (0.91–1.44) | 0.22 |
| IgH negative | 0.54 | 1.15 (0.81–1.62) | 0.38 |
| IgH positive and t11;14 negative | 0.54 | 1.14 (0.84–1.54) | 0.41 |

[a]—frequency of risk allele G; [b]—OR odds ratio, CI confidence interval.

### 3.1. Clinical Phenotypes of AL

We used the same GWAS data to assay for any genetic distinctions within AL with different clinical presentation [30]. A total of nine clinical profiles were selected among organ involvement (kidney, heart, heart, and kidney and liver, irrespective of whether other organs were involved) and Ig profiles (intact IgG with λ or κ, λ any, κ any, λ/κ light chain only (LCO), and λ LCO). Patient numbers varied from the most common, λ any (with or without heavy chains) with 930 patients to the smallest one of liver profile with only 194 patients (Table 2). Among Ig profiles, the λ/κ LCO profile showed a strong association with SNP rs9344, OR 1.62 (*p* = $1.99 \times 10^{-12}$) and in the λ LCO profile it was 1.70 (*p* = $1.29 \times 10^{-11}$). The IgG profile with isotype IgG λ showed OR of 1.57 (*p* = $2.90 \times 10^{-8}$). Notably, while LCO profiles for AL were highly significant (p~$10^{-11}$) for cyclin D1 splice site SNP rs9344, LCO profiles for MM were not increased (OR 1.01/1.03) (Table 2). SNP rs6752376 marked the heart and kidney profile (OR 1.54, *p* = $2.88 \times 10^{-8}$) and SNP rs7820212 marked the liver profile (OR 1.86, *p* = $1.86 \times 10^{-8}$). We carried out bioinformatic analysis into the possible functional bases of the observed associations, but the mechanisms for all but rs9344 remained speculative [30]. Manhattan plots are shown for joint analysis in four clinical profiles with genome-wide associations (Figure 3). The figures illustrate the distinction between the profiles: the peaks reaching the genome-wide association (red line) are only observed for individual profiles. While the LCO profile was strongly associated with rs9344, the weakest association was noted for the IgG profile. Conversely, rs10507419 was strongly associated only with the IgG profile.

**Table 2.** Association of risk allele G in SNP rs9344 with light chain only AL (reference [26]).

| Profiles | Number of Cases | OR | 95% CI [a] | *p*-Value [b] | *I²* [c] | Z-Score |
|---|---|---|---|---|---|---|
| Overall AL | 1129 | 1.35 | 1.23–1.48 | $7.80 \times 10^{-11}$ | 0.36 | 6.51 |
| IgG | 447 | 1.20 | 1.05–1.38 | $9.69 \times 10^{-3}$ | 0.00 | 2.59 |
| λ any | 930 | 1.40 | 1.27–1.55 | $9.28 \times 10^{-11}$ | 0.00 | 6.48 |
| κ any | 265 | 1.33 | 1.11–1.59 | $2.03 \times 10^{-3}$ | 0.00 | 3.09 |
| λ/κ LCO | 535 | 1.62 | 1.42–1.85 | $1.99 \times 10^{-12}$ | 0.00 | 7.04 |
| λ LCO | 404 | 1.70 | 1.46–1.98 | $1.29 \times 10^{-11}$ | 0.00 | 6.77 |
| Kidney | 844 | 1.34 | 1.20–1.48 | $6.89 \times 10^{-8}$ | 0.20 | 5.40 |
| Heart | 835 | 1.39 | 1.24–1.54 | $2.91 \times 10^{-9}$ | 0.49 | 5.94 |
| HK | 426 | 1.31 | 1.14–1.52 | $2.14 \times 10^{-4}$ | 0.38 | 3.70 |
| Liver | 194 | 1.40 | 1.14–1.73 | $1.63 \times 10^{-3}$ | 0.00 | 3.15 |
| Overall MM | 3790 | 1.06 | 1.00–1.12 | $4.00 \times 10^{-2}$ | | |
| MM LCO κ [d] | 123 | 1.01 | 0.78–1.30 | 0.95 | 0.61 | 2.09 |
| MM LCO λ [d] | 89 | 1.03 | 0.75–1.30 | 0.87 | | |

[a] CI, confidence interval; [b] *p*-value based on the meta-analysis of the three patient cohorts in AL amyloidosis, and two patient cohorts in multiple myeloma; [c] *I²* proportion of total variance due to heterogeneity; [d] based on German MM data. Genome-wide significant associations are indicated in bold.

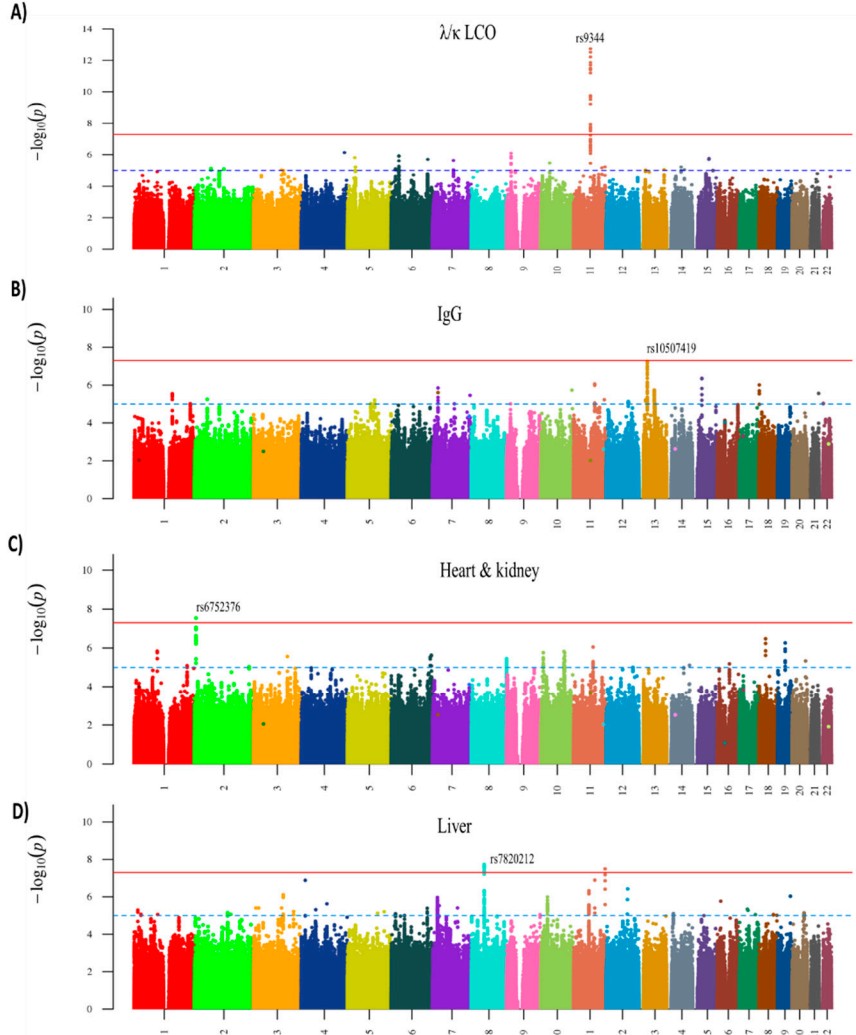

**Figure 3.** Manhattan plots of association analysis for AL amyloidosis clinical profiles with genome-wide significant results. (**A**) λ/κ LCO profile; (**B**) IgG profile; (**C**) heart and kidney profile; (**D**) liver profile. The x-axis shows the chromosomal position and the y-axis is the significance ($-\log_{10}$ P; 2-tailed) of association derived by logistic regression. The red line shows the genome-wide significance level ($5 \times 10^{-8}$) and the blue line shows a suggestive significance level ($1 \times 10^{-5}$). The top SNP of each significant association is labeled. From reference [30] with permission.

We could not address the individual role of LCO and t(11;14), because in our dataset only the German cohort had data on t(11;14). Availability of such data could have resolved the question, if the preferential association of rs9344 with LCO profiles could be explained by the association of this SNP with t(11;14) and the resulting disturbance of IgH production in AL and MM [25,26,31]. However, no light chain excess was reported in t(11;14) AL [32,33]. Data on an MM cell line suggested that compromised production of IgH leads to excess production of free light chains [34]. An alternative explanation could be that rs9344 interferes with IgH production independent of t(11;14). Risk allele G at the splice site of cyclin D1 encodes the full length cyclin D1, which has many functions, including involvement in double-strand repair with RAD51, BRCA1, and BRCA2 and, thus, a possible interference with the class switch recombination for IgH may be plausible [35,36].

The results were also remarkable in that the associations were distinct from those found for all AL, even though the sample size in each profile was smaller [26]. This may indicate that the profiles were able to define AL into molecular subtypes with increasing genetic homogeneity and with possible therapeutic implications. Particularly striking were the distinctly non-overlapping genetic associations for the LCO and IgG isotypes.

For rs9344, the preference for LCO AL is a lead that should provide insight into the underlying mechanisms.

### 3.2. Combined Analysis of AL, MM and MGUS

By 2018, GWASs had identified 23 risk loci for MM [37–40]. Two separate GWASs were done on MGUS, and the MGUS and AL results were compared with the results from MM [41,42]. To explore the possibility of shared genetic loci, we carried out a joint analysis combining GWAS data on 4403 MM, 992 MGUS, and 1230 AL amyloidosis cases and 10,554 controls [43]. In this analysis, 17 loci reached a genome-wide significance of $5 \times 10^{-8}$, including eight previously unreported loci. Functional annotation suggested that novel pathways might be involved. Some of these mapped close to genes possibly related to hematopoiesis, including ASXL2 encoding a Polycomb group protein, which interacts with tumor suppressor BAP1. Another one was ARHGAP26, which is a fusion partner with MLL in myelodysplastic syndrome. Further loci were HLA-DRA at a locus regulating expression of HLA-DQ and HLA-DR, and TPTE2 encoding a membrane-associated protein tyrosine phosphatase, homolog of PTEN. The 17 SNPs were enriched at histone marks and transcription factor binding sites, and the associated genes were highly expressed in precursor B lymphoid cells.

## 4. Mendelian Randomization (MR)

The rarity of AL represents a barrier to the identification of risk factors for the disease through conventional epidemiological observational studies. MR is an analytical method that exploits genetic variants as instrumental variables (IVs) to infer the causal relevance of an exposure to disease risk. Because the genetic variants are randomly assigned at conception, they are not influenced by reverse causation, and in the absence of pleiotropy (i.e., genetic variants being associated with a disease through alternative pathways), they can provide unconfounded estimates of disease risk (Figure 4). GWASs have identified associations between SNPs and large numbers of traits/phenotypes, offering the prospect of identifying causal relationships for diseases, such as AL, through MR-based analyses.

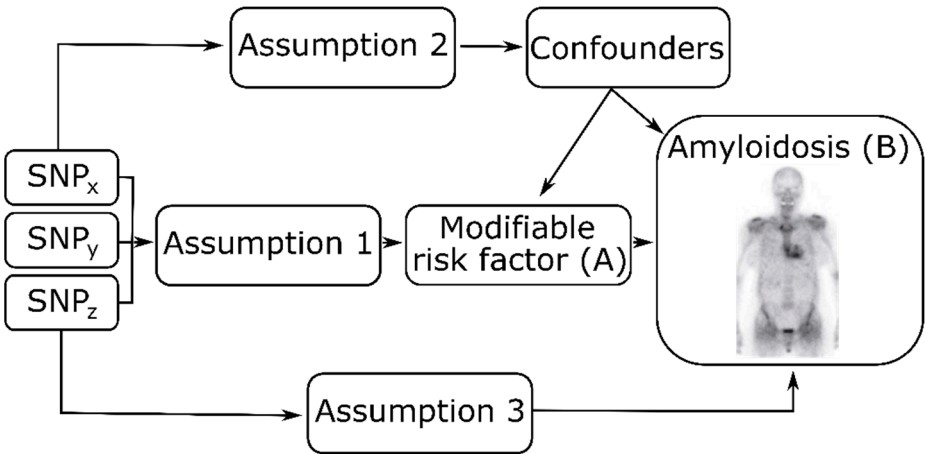

**Figure 4.** Principles of Mendelian randomization analyses and assumptions that need to be satisfied for unbiased analyses. Assumption 1 indicates genetic variants used as instrumental variables (i.e., SNPs) are only associated with the modifiable risk factor. Assumption 2 indicates genetic variants are not associated with any measured or unmeasured confounders. Assumption 3 indicates genetic variants only influence the risk of developing AL amyloidosis through the modifiable risk factor. Courtesy of Charlie Saunders.

The first MR was based on the above GWAS combining the three dyscrasias [43]. The environmental factors (exposures) included phenotypes possibly linked to plasma cell dyscrasias, including serum albumin and creatinine levels, bone mineral density,

hemoglobin, and immunoglobulin (Ig) traits, the latter obtained from healthy individuals [44]. The strongest signal was observed for IgA: MGUS (OR = 5.16, $p = 2.9 \times 10^{-8}$), MM (4.11, $2.6 \times 10^{-19}$), AL (2.54, $7.9 \times 10^{-4}$). Among other Ig isotopes, IgM showed weaker, but significant associations for all three diseases [45]. No associations were found for bone mineral density, hemoglobin, serum albumin, or creatinine level.

In a separate study on AL, we used the above GWAS data, but a wider repertoire of phenotypes [46]. However, to maximize our prospects of identifying a causal relationship with AL we restricted the MR analysis by considering 72 phenotypes with a possible mechanistic link to AL [2,43,46]. Associations with Bonferroni-defined significance level were observed for genetically predicted increased monocyte counts ($p = 3.8 \times 10^{-4}$) and the tumor necrosis factor receptor superfamily member 17 (TNFRSF17, alias BCMA) ($p = 3.4 \times 10^{-5}$). Two other associations with the TNFRSF (members 6, alias Fas, and 19L, alias RELT) reached a nominal significance level. Another unrelated association was observed between genetically predicted decreased fibrinogen levels and AL, which may be related to non-hemostatic roles of fibrinogen.

MR results do not explain mechanisms and, in this case, they were speculative [45]. It is plausible that a causal relationship with monocyte concentration could be explained by selection of a light chain-producing clone during progression of MGUS towards AL. This could be influenced by macrophages and dendritic cells for which monocytes are precursors. MM bone marrow is rich in macrophages, which are important components in the tumor microenvironment [47,48]. Macrophages are known to contribute to immune suppressive bone marrow microenvironment in MGUS and MM [49]. One could speculate that high monocyte/macrophage levels impose a selective pressure on MGUS bone marrow, facilitating expansion of a light chain-producing clone.

The TNF superfamily is large with 19 known ligands and 29 receptors with specific and diverse functions [29]. These exhibit generally pro-inflammatory properties through the activation of NF-kappaB pathway [29]. Two of the present receptors mediate apoptotic signaling with apparently opposite directions in the present study, TNFRSF6 increased and RELT decreased the risk of AL. However, the effect size (OR) of RELT was weak compared to that of TNFRSF17 (BCMA), which may be a positive disease driver. Since TNFRSF proteins have key functions in lymphocyte biology, it is plausible that they offer a potential link to AL pathophysiology.

There are large amounts of relevant biological data on TNFRSF17 (BCMA). It is preferentially expressed in mature B-lymphocytes, and is important for B-cell development and autoimmune response. Ligands for TNFRSF17 include B-cell activating factor (BAFF) and a proliferation-inducing ligand (APRIL), leading to NF-kappaB and MAPK8/JNK activation [50]. A closely related analog, TNFRSF13B was found to be a low-risk risk gene in one of our early GWAS in MM, and it reached genome-wide significance also in the above pooled study of MM-AL-MGUS [38,43]. TNFRSF17 is overexpressed in MM and AL patients, and is being tested as a therapeutic target for AL [51,52]. The rationale in many novel immunotherapy approaches in MM has been to use TNFRSF17 (BCMA) as an immunological target on homing on MM cells and combining in the same drug the TNFRSF17 antibody with another antibody attracting effector T-cells, such as CD3+ [50,53]. The approach is being applied in three modalities of treatment, including antibody-drug conjugates (ADCs), bispecific T-cell engagers (BITEs), and chimeric antigen receptor (CAR) T-cell therapies [54].

## 5. Conclusions

Epidemiology of AL and other non-hereditary types of amyloidosis is a moving target, for several reasons. Diagnostics are not uniform at the population level for any of these diseases. For AA amyloidosis, the varying spectrum of chronic inflammatory conditions and their control, and for ATTR, the aging population all contribute to changes in incidence estimates and their time trends. For the same reasons, accurate population level survival estimates will be difficult to obtain.

Our genetic studies highlighted the importance of the SNP at the cyclin D1 splice site influencing t(11;14) in AL and MM. Overall t(11;14) is more prevalent in AL compared to MM, and the influence of the cyclin D1 splice site SNP was far more prominent in AL compared to MM. For this translocation in MM, promising treatment was achieved with a novel drug venetoclax [55,56]. As could be predicted from the key role of t(11;14) in AL, this drug has now shown excellent efficiency in AL [57]. Our other finding, which touched on mainstream therapeutic drug design, was the observed association of TNFRSF proteins with AL risk in the MR study, notably with TNFRSF17 (BCMA), which has been the main target of MM-directed immunotherapy [54]. BMCA-targeted CAR T-cell therapy showed clinical activity in MM patients with relapsed and refractory disease [58]. It is likely a question of time when it is validated in AL.

The GWAS approach in targeting low-risk genetic loci has sometimes been criticized as not being able to deliver clinically relevant results. While we do not want to enter this debate here, we just refer to the above findings and conclude that GWAS has described the germline genetic landscape of AL, and has been able to genetically decipher therapeutic targets, which are gaining clinical use in MM and AL.

**Author Contributions:** K.H.; writing, A.F. reviewing, both agreed to submission. Both authors have read and agreed to the published version of the manuscript.

**Funding:** K.H. was supported by the European Union's Horizon 2020 research and innovation program, grant no 856620. A.F. was supported by the German Jose Carreras Leukemia Foundation.

**Institutional Review Board Statement:** Not applicable.

**Informed Consent Statement:** Not applicable.

**Data Availability Statement:** Not applicable.

**Conflicts of Interest:** The authors declare no conflict of interest.

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
