# Peer review of "Epidemiology of Amyloidosis and Genetic Pathways to Diagnosis and Typing"

_hemato, doi:10.3390/hemato2030027_

Round 1

Reviewer 1 Report

The authors reviewed articles regarding amyloidosis by focusing on ATTR and AL amyloidoses. Issues regarding epidemiology and genetic characteristics are comprehensively summarized.

This is an interesting review article. Taking up the topic of systemic amyloidosis is timely because novel therapeutic options for this disease, such as therapeutic agents against plasma cell dyscrasia, transthyretin stabilizers, small interfering RNA, and antisense oligonucleotide, now appear one after another. I do not have any critical comments.

Minor issues and suggestions to strengthen this manuscript are raised as follows: 

  1. The number of affiliation for “Division of Pediatric Neurooncology, German Cancer Research Center (DKFZ), German Cancer Consortium 10 (DKTK)” would be “4”, but not “3”.
  2. Please reconfirm the use of abbreviations throughout the manuscript. For example, “MM” in the abstract should be spelled out at its first appearance.
  3. According to the nomenclature recommendation (Amyloid 2020; 27: 217-222), “hereditary ATTR amyloidosis” and “wild-type ATTR amyloidosis” are abbreviated as “ATTRv (v for variant)” and “ATTRwt”, respectively. I would recommend using these terms.
  4. Patients with ATTRv amyloidosis were initially reported in endemic foci of Portugal, Japan, and Sweden, whereas recent studies revealed that late-onset ATTRv amyloidosis patients were prevalent even in non-endemic areas (Neurol Ther 2020; 9: 317-333). This issue should be incorporated, by citing this article.

Author Response

Thanks, all suggestions implemented.

Reviewer 2 Report

In this paper, the authors present a review of their own work in the epidemiology and genetics amyloidosis, mainly in Sweden.

This topic is interesting and invites you to read; however, the title is misleading: the reader is expecting to find a comprehensive review of all the research on the topic. However, the authors limit themselves to present their own work without relation to other research, mainly from data in Sweden. This should be presented in the title.

As the authors mention, the data in the field is limited, and although the study is well written and organized, it is only a list of the work by the research group that does not improve our understanding of the topic.

In the epidemiology section, it is not clear how the cancer risk paragraph is related to the epidemiology of amylodosis. The conclusion/explanation of Figure 1 are not clear.

In the section “Genetics of the germline of AL”, what conclusion can be reached with the fact that: The magnitude of the association was similar in AL and MM even though the translocation is 3 times more prevalent in AL?

Figure 4 is not clear.

Author Response

Reviewer 2 expected a comprehensive review. We had accepted the title that was recommended to us. In the epidemiology section we believe all representative (population-based) studies are referred to along our Swedish studies. In the genetics part, we covered all literature on germline genetics. We do not agree with his point about Fig. 4 (similar figures are shown in other papers, explain Mendelian randomization, including the new reference 43). We covered all his specific point.

Reviewer 3 Report

the paper is well written and correct!  supported by an excellent bibliography.  consistent in conclusions

Author Response

Thanks!